# CD138 Is Expressed in Different Entities of Salivary Gland Cancer and Their Lymph Node Metastases and Therefore Represents a Potential Therapeutic Target

**DOI:** 10.3390/ijms23169037

**Published:** 2022-08-12

**Authors:** Marcel Mayer, Lisa Nachtsheim, Franziska Hoffmann, Ferdinand von Eggeling, Orlando Guntinas-Lichius, Johanna Prinz, Jens Peter Klußmann, Alexander Quaas, Christoph Arolt, Philipp Wolber

**Affiliations:** 1Department of Otorhinolaryngology, Head and Neck Surgery, Medical Faculty, University of Cologne, 50931 Cologne, Germany; 2Department of Otorhinolaryngology, Head and Neck Surgery, Jena University Hospital, Friedrich-Schiller-University, 07747 Jena, Germany; 3Department of Otorhinolaryngology, MALDI Imaging and Innovative Biophotonics, Jena University Hospital, Friedrich-Schiller-University, 07743 Jena, Germany; 4Department I of Internal Medicine, Center for Integrated Oncology Aachen Bonn Cologne Duesseldorf, University of Cologne, 50937 Cologne, Germany; 5Medical Faculty, Institute of Pathology, University of Cologne, 50937 Cologne, Germany

**Keywords:** salivary gland neoplasm, immunohistochemistry, head and neck cancer, targeted therapy, CD138, Syndecan-1, MALDI-MS imaging

## Abstract

Advanced salivary gland carcinomas (SGC) often lack therapeutic options. Agents targeting CD138 have recently shown promising results in clinical trials for multiple myeloma and a preclinical trial for triple-negative breast cancer. Immunohistochemistry for CD138 was performed for all patients who had undergone primary surgery for SGC with curative intent. Findings were validated using matrix-assisted laser desorption/ionization mass spectrometry (MALDI-MS) imaging. Overall, 111 primary SGC and 13 lymph node metastases from salivary duct carcinomas (SaDu) were evaluated. CD138 expression was found in 60% of all SGC with differing expression across entities (*p* < 0.01). A mean of 25.2% of the tumor cells in mucoepidermoid carcinoma (MuEp) were positive, followed by epithelial-myoepithelial carcinoma (20.9%), acinic cell carcinoma (16.0%), and SaDu (15.2%). High-/intermediate-grade MuEp showed CD138 expression in a mean of 34.8% of tumor cells. For SaDu, lymph node metastases showed CD138 expression in a mean of 31.2% of tumor cells which correlated with CD138 expression in their primaries (*p* = 0.01; Spearman’s ρ = 0.71). MALDI-MS imaging confirmed the presence of the CD138 protein in SGC. No significant association was found between clinicopathological data, including progression-free survival (*p* = 0.50) and CD138 expression. CD138 is expressed in the cell membrane of different entities of SGC and SaDu lymph node metastases and therefore represents a potential target for CD138 targeting drugs.

## 1. Introduction

Salivary gland carcinomas (SGC) are rare, as they represent approximately 6% of all malignant head and neck tumors [1]. About 70% of SGC originate from the parotid gland, followed by 20% from the submandibular gland. The remaining SGC are located in the sublingual gland and the minor salivary glands [2]. SGC show a marked histopathological heterogeneity with 22 entities described according to the current WHO classification [3]. Aggressiveness and long-term prognosis differ not only when comparing different entities, but also within entities depending on tumor characteristics such as grading. Accordingly, the median 5-year overall survival of low-grade mucoepidermoid carcinoma (MuEp) has been shown to be as high as 92–100%, whereas the median 5-year overall survival of high-grade MuEp and salivary duct carcinoma (SaDu) is ranging around 40%, respectively [4,5].

The recommended therapy for resectable SGC without distant metastasis is surgical removal of the primary tumor. Ipsilateral neck dissection is recommended in case of suspected loco-regional lymph node involvement, high-grade malignancy, or advanced stage (T3–T4) tumor [6]. Postoperative radiation therapy should be performed in case of adenoid cystic carcinoma (ACC) histology, positive margins, high-grade malignancy, perineural invasion, or advanced stage (T3–T4) tumor [6]. In the presence of recurrent and unresectable or metastatic disease, platinum-based palliative chemotherapy regimens have been mainly used during the last decades, with response rates lower than 30% associated with considerable toxicity [7,8,9]. Consequently, different molecular targets have been identified in recent years such as HER-2, androgen receptor, the NTRK fusion protein, EGFR, VEGF, and Trop-2 for a molecularly targeted therapy approach with partially promising clinical data in selected patients [8,10,11]. However, in many patients with advanced SGC therapeutic targets are still missing, leading to limited therapeutic options in these cases. Therefore, the identification of further targets in SGC is of utmost importance, particularly in aggressive entities with an unfavorable long-term outcome such as high-grade MuEp, SaDu, and ACC [4,5,12].

CD138, also known as Syndecan-1, is a type 1 transmembrane heparan sulfate proteoglycan belonging to the Syndecan family [13]. Consisting of 288 amino acids, it includes a cytoplasmatic C-terminal domain, an extracellular N-terminal domain possessing various glycosaminoglycan chains, and a transmembrane region [14]. CD138 is physiologically expressed on epithelial cells and leukocytes [14]. Previous studies have shown that CD138 interacts with different growth factors, promotes cell migration, modulates inflammatory processes, and interacts with the extracellular matrix [15,16,17,18]. Particularly, it acts as a co-receptor, binding VEGF, Wnt, and FGF, and resulting in activation of downstream intracellular signaling pathways [19]. It further plays a role in downstream activation of PI3K/Akt and Ras/MAPK pathways, regulating the cell cycle and cell proliferation [20].

CD138 expression, assessed by IHC, was previously found in non-cancerous salivary gland tissue. More precisely, a weak CD138 staining was found in ductal, myoepithelial, intralobular duct, and some serous cells, whereas most serous cells showed a moderate staining pattern [21]. Interestingly, a recently published study showed an upregulation of CD138 expression in ductal epithelial salivary gland cells of patients with Sjögren’s syndrome (SS) compared to the expression in ductal epithelial cells of healthy patients. Further, the study found significantly higher salivary CD138 levels in SS patients compared to healthy patients and a positive correlation between plasma CD138 levels and SS disease activity, suggesting CD138 as a potential biomarker for disease activity in SS [22].

Presently, CD138 has the highest clinical importance in multiple myeloma (MM) as it is highly expressed in differentiated plasma cells and MM cells and is therefore used as a marker for the identification of MM cells [23]. Additionally, CD138 is a prognostic marker in MM as high serum levels of shed CD138 correlate with poor prognosis [13]. In a preclinical study, the antibody-drug conjugate indatuximab ravtansine (BT062) showed complete remission, i.e., >95% reduction of median tumor volume in relation to controls, in triple-negative breast cancer xenografts with strong expression of CD138 in IHC [24].

As advanced SGC often lack sufficient therapeutic options and as more drugs targeting CD138 emerge, it was the aim of this study to investigate the expression of CD138 in various entities of SGC using a bimodal approach of immunohistochemistry (IHC) and matrix-assisted laser desorption/ionization mass spectrometry (MALDI-MS) imaging.

## 2. Results

### 2.1. Patients’ Cohort

Overall, 111 patients with primary SGC of the parotid gland (90.1%) and the submandibular gland (9.9%) were included. Gender was equally distributed (females: 51.4%; males: 48.6%,) and the mean age was 55.7 years (±17.6). The most frequent entity was SaDu (24.3%, *n* = 27), followed by MuEp (22.5%, *n* = 25) and ACC (20.7%, *n* = 23). Other rare entities (OTH, *n* = 9) were four basal cell carcinomas (BCC; 3.6%), three myoepithelial carcinomas (2.7%), one carcinosarcoma (0.9%), and one polymorphous adenocarcinoma (0.9%). Fifty-five patients (49.5%) showed pathological T-stage 1–2, while 53 patients (47.7%) showed advanced pathological T-stage 3–4. Thirty-three patients (30.6%) had pathological loco-regional lymph node metastasis. Further demographic and histopathological data are displayed in Table 1.

### 2.2. Immunohistochemistry

The most frequent expression of CD138 was found in MuEp with 25.2% (±26.7) of CD138 positive tumor cells, followed by EpMy with 20.9% (±28.9), Acin with 16.0% (±24.9), and SaDu with 15.2% (±22.0). A generally rare CD138 expression was found in ACC with 3.0% (±7.1) and SeC with 0.5% (±1.4). Across all entities, a mean of 13.9% (±21.8) of tumor cells showed CD138 expression (Table 1). Interestingly, high-/intermediate-grade MuEp (*n* = 10/25) showed a particularly frequent CD138 expression with 34.8% (±32.6) of tumor cells expressing CD138. The immunohistochemical protein expression of CD138 is shown for exemplary cases in Figure 1. The distribution of CD138 expression among the most frequent entities is displayed in Figure 2. A statistically significant difference between the different entities in terms of CD138 expression was found (*p* < 0.01). Figure 3 shows the homogeneity of CD138 expression across the four TMAs per case among the most frequent entities as standard deviation. The mean standard deviation of CD138 expression across the four TMAs per case was 8.4%.

Besides the primaries, CD138 expression was studied in 13 loco-regional lymph node metastases originating from SaDu. A markedly more frequent CD138 expression with a mean of 31.2% (±35.4) of tumor cells being positive for CD138 was found in the SaDu lymph node metastases compared to the SaDu primaries (15.2% (±22.0)) (Figure 4). For SaDu, CD138 expression in primaries correlated with expression in lymph node metastases (*p* = 0.01; Spearman’s ρ = 0.71) (Figure 5). 

CD138 IHC of full sections from five cases with high intratumoral staining heterogeneity (3 SaDu, 2 MuEp) confirmed the initial findings. Two SaDu displayed a true heterogeneous expression. The first was overall poorly differentiated. The other stained positive in well differentiated areas. Whilst poorly differentiated, solid areas were mostly negative. One SaDu displayed a reduced staining intensity in the centre of the FFPE slide. Intracystic epithelia in MuEp that appeared flattened—possibly due to intracystid fluid pressure—did not stain for CD138, whereas most intact mucinous and epitheloid cells in two analyzed MuEp cases were positive.

CD138 IHC of two full sections from non-cancerous salivary gland tissue revealed partial, moderate membranous expression in excretory and striated ducts and strong membranous expression in intercalated ducts, whereas the acini were negative for CD138.

### 2.3. MALDI-MS Imaging

MALDI-MS imaging was performed for validation of the immunohistochemical data for 8 exemplary cases of primary SGC (Figure 6). One mass (1767.001 *m*/*z*; ((K)EGEAVVLPEVEPGLTAR(E); M+H^+^) could be correlated to a peptide corresponding to CD138. The results reveal that CD138 was detected in all 8 samples. One SeC with missing membranous CD138 expression in IHC showed the highest intensity of CD138 expression in MALDI-MS imaging within the analyzed cores. Among the other CD138 positive cases in IHC, the highest intensities in MALDI-MS imaging were found for one MuEp with CD138 positivity of 20% of the tumor cells in IHC, followed by one ACC with CD138 positivity of 15% of the tumor cells in IHC, and one MuEp with CD138 positivity of 47.5% of the tumor cells in IHC.

Table 2 displays the association between the immunohistochemical CD138 expression (yes vs. no), tumor localization, demographic, and histopathological data. No significant association was found between CD138 and tumor localization, sex, T-stage, N-stage, perineural invasion, (lympho-)vascular invasion, extracapsular extension, or grading among all entities, and when tested for the individual entities.

### 2.4. Survival

The 5-year progression-free survival (PFS) among all entities was 70.3% (78 out of 111). The mean follow-up was 67.2 (±50.8) months. Twenty patients showed tumor recurrence (9 with loco-regional recurrence, 11 with distant metastasis) within five years after the first diagnosis. The lowest PFS was found for SaDu with 48.1% (4 patients with loco-regional recurrence, 5 with distant metastasis). SeC had the highest PFS with 100.0%. PFS was 72.7% for patients with CD138 expression and 66.7% for patients without CD138 expression (Figure 7), which showed no statically significant difference (*p* = 0.71). Moreover, PFS did not significantly differ between patients with and without CD138 expression in the subgroups MuEp (*p* = 0.81), EpMy (0.56), Acin (*p* = 0.81), SaDu (*p* = 0.62), ACC (*p* = 0.54), OTH (*p* = 0.17), and high-/intermediate-grade MuEp (*p* = 0.94).

## 3. Discussion

The aim of this study was to evaluate the expression of CD138 and its correlation with clinicopathological data, as well as survival in a large cohort of different entities of SGC, using a bimodal approach of immunohistochemistry and MALDI-MS imaging. 

The most frequent expression of CD138 was found in MuEp, followed by EpMy, Acin, and SaDu. Interestingly, considering high-/intermediate-grade MuEp, this subgroup showed an even more frequent CD138 expression compared to low grade tumors of the same entity. The frequent expression of CD138 in high-/intermediate-grade MuEp, and the considerable expression of CD138 in SaDu are also of special interest, as these entities show a particularly unfavorable prognosis [4,5] and therefore urgently necessitate additional molecular targeted therapies. To date, there is only one study revealing the expression of CD138 in a limited number of three entities of SGC. In this study by Alaeddini et al., 7 out of 9 Acin showed an epithelial CD138 expression in >10% of the tumor cells, whereas 11 out of 30 ACC, and 7 out of 30 MuEp expressed CD138 in >10% of the tumor cells [25]. The reported more frequent expression of CD138 in Acin and ACC compared to the results of the present study is most likely due to the fact that the authors measured CD138 expression as cumulative cytoplasmatic and membranous expression, whereas the present study solely assessed the membranous CD138 expression. Assessing the membranous expression of a potential molecular target of the tumor cell seems plausible as the target is located in the cell membrane and is in line with immunohistochemical assessment of other molecular targets, as e.g., HER2 in breast cancer [26].

Besides the sole extent of expression of a target, its homogeneity of expression within the tumor plays a role for the effectiveness of an antibody-drug conjugate. Precisely, advanced gastric/gastroesophageal junction cancer patients with homogenous HER2 staining patterns were reported to have a longer median overall survival compared to those with heterogenous and focal expression when treated with trastuzumab emtansine [27]. To address this issue, the CD138 expression was analyzed in four TMAs per patient in the present study. A mean standard deviation of 8.4% across the four TMAs per case displays a generally homogenous CD138 expression pattern. Therefore, it can be assumed that a biopsy from an SGC primary or metastasis can provide a representative snapshot of CD138 expression within the whole tumor.

The assessment of the expression of a potential molecular target seems not only of importance in the primary tumor, but also in loco-regional and distant metastases, as these lesions are targeted in case of a systemic therapy. The present study showed a markedly higher CD138 expression in lymph node metastases originating from SaDu tumors than in their primaries. This finding must be emphasized as previous studies solely assessed CD138 expression in primary tumors [28], although its expression in metastatic lesions is of great importance when targeting advanced tumors.

The results of IHC were validated using MALDI-MS imaging. The CD138 protein was detected in all eight cases of primary SGC. Among the seven CD138 positive cases in IHC, the highest intensities in MALDI-MS imaging were found for one MuEp with CD138 positivity of 20% of the tumor cells in IHC, followed by one ACC with CD138 positivity of 15% of the tumor cells in IHC, and one MuEp with CD138 positivity of 47.5% of the tumor cells in IHC. Interestingly, one SeC with missing CD138 expression in IHC showed positive CD138 protein expression in MALDI-MS imaging. The most likely explanation is that MALDI-MS Imaging analyzed the CD138 protein in the whole TMA cores, including the stromal, membranous, and cytoplasmatic CD138 protein expression of tumorous and adjacent non-tumorous tissue, whereas the immunohistochemical CD138 expression was exclusively assessed in the membrane of the tumor cells.

As mentioned above, various agents targeting CD138 have been developed in recent years. One of these is VIS832, a humanized IgG1-κ mAb with high binding avidity to membranous CD138, leading to immune cell-mediated cytotoxicity [29]. A shortly published preclinical trial showed effectiveness of VIS832 in vivo and in vitro. Precisely, a VIS832-induced dose-dependent cytolysis of autologous CD138 positive MM cell lines as well as a marked survival benefit in mice treated with VIS832 compared to no therapy after injection of CD138 positive MM cells was reported [29]. A second and even more promising agent is the antibody-drug conjugate indatuximab ravtansine, consisting of the CD138-specific antibody BB4 conjugated to the cytotoxic maytansinoid drug DM4 [30]. In a clinical phase 1/2a study, indatuximab ravtansine showed an objective response rate of >70% and a clinical benefit rate of >85% in combination with dexamethasone and lenalidomide or pomalidomide in a cohort of patients with refractory MM after chemotherapy [31]. As CD138 is also expressed in normal epithelia, leukocytes, and, as shown in this study, in non-cancerous salivary gland tissue, unwanted side-effects of indatuximab ravtansine have to be considered [28]. The most frequent adverse effects are grade 1 or 2 diarrhea, fatigue, or nausea [32]. Possibly, but less frequently, grade 3 or 4 adverse effects are anemia, thrombocytopenia, and neutropenia [31]. However, the drug seems to be generally well tolerated. Although CD138 is expressed in salivary gland tissue, salivary gland specific side effects such as dry mouth or sialadenitis have not been reported yet [31,32]. Additionally, the effectiveness of indatuximab ravtansine has recently preclinically been studied in solid tumor cells in vitro and in vivo. Triple-negative breast cancer cell lines with higher CD138 expression, evaluated with fluorescence-activated cell sorting (FACS), were markedly more sensitive to indatuximab ravtansine than triple-negative breast cancer cell lines with two-fold lower CD138 expression. Moreover, in a triple-negative breast cancer xenograft mouse model with strong membranous CD138 expression (>75% of positive cells), assessed by IHC, indatuximab ravtansine was highly effective, leading to strong inhibition of tumor growth and complete remission (>95% reduction of median tumor volume in relation to controls), while the effect on tumor inhibition was markedly lower but still detectable in a triple-negative breast cancer xenograft with weak CD138 expression (<25% of positive cells) [24]. These results suggest a positive correlation between CD138 expression in IHC and response to indatuximab ravtansine. To date, there is no preclinical or clinical study evaluating the effectiveness of indatuximab ravtansine or VIS832 in SGC. Although a mean of only 13.9% of tumor cells were positive for CD138 across all entities, certain subgroups such as MuEp, high-/intermediate MuEp, and SaDu lymph node metastases showed a CD138 expression in 25.2%, 34.8%, and 31.2% of tumor cells. Moreover, particular cases of MuEp, Acin, SaDu, and SaDu lymph node metastases revealed a CD138 expression in 80% or more of their tumor cells. Against the background of few effective treatment options in advanced stage SGC, patients with tumors having a particularly frequent CD138 expression could benefit from CD138 targeting drugs. 

The results of the present study show no significant association between CD138 expression and localization of the primary, sex, T-stage, presence of lymph node metastasis, perineural invasion, (lympho-)vascular invasion, extracapsular extension, or grading for the whole cohort, as well as for the individual entities. This is in line with the abovementioned study by Alaeddini et al., who found no significant correlation between localization, sex, and presence of lymph node metastases and CD138 expression in MuEp and ACC [25]. It is worth mentioning that an association between a higher expression of CD138 and lower T- and N-stages has been found for squamous cell carcinoma of the head and neck (HNSCC) [33,34], which cannot be confirmed for SGC against the background of the present study.

High expression of membranous CD138 has been shown to be associated with a favorable outcome in various malignancies such as HNSCC, mesothelioma, gastric, and hepatocellular cancer, and with an unfavorable outcome in other malignancies such as pancreatic, ovarian, and thyroid cancer [19,28]. The present study is the first to investigate the CD138 expression as a prognostic marker in SGC. No statistically significant association between CD138 expression and PFS for the whole cohort and for the entities MuEp, EpMy, Acin, SaDu, or ACC was found.

When interpreting the results of this study, the retrospective collection of clinicopathological data must be considered as a limitation as well as the limited numbers of cases per entity, which is owed to the generally low incidence of SGC.

Overall, this study gives a comprehensive overview of the membranous and extracellular expression of CD138 in various entities of SGC. The results reveal that CD138 is frequently expressed in the membrane of particular entities of primary SGC and their metastases. Therefore, preclinical studies evaluating the effectiveness of, e.g., indatuximab ravtansine in SGC, especially in entities with frequent CD138 expression in IHC, should be performed to provide patients with advanced SGC with further tailored therapeutic options in the future.

## 4. Methods and Materials

### 4.1. Cohort

Patients were included in this analysis if they had undergone primary surgery with curative intent for primary SGC of the parotid or submandibular gland at the Department of Otorhinolaryngology, Head and Neck Surgery, University of Cologne between 1990 and 2019, and if sufficient formalin-fixed paraffin-embedded material of the primary tumor was available. Demographic, histopathological, and survival data were retrieved from clinical records and histopathological reports with respect to tumor characteristics including the stage of disease at the time of diagnosis according to the AJCC TNM staging system (8th edition, 2020) [35]. In case of missing data on follow-up, patients or their general practitioners were phoned to follow up in terms of the current tumor status. A follow-up until progression or death within 60 months or until May 2022 in case of initial surgical treatment after May 2017 or of at least 60 months in case of initial surgical treatment before May 2017 was available for all patients included in this study. The study was performed according to the regulations of the Ethics Committee of the University of Cologne.

### 4.2. TMA Preparation and Immunohistochemical Assessment of CD138 Expression

Four tissue cylinders per case with a diameter of 1.2 mm were punched out from one tumor-bearing formalin-fixed, paraffin-embedded (FFPE) block using a semiautomated precision instrument. The cylinders were then transferred to an empty paraffin block. Overall, 496 tissue microarrays (TMA) represented 111 cases of primary SGC and 13 SaDu lymph node metastases. SaDu lymph node metastases were included, as this entity shows a particularly aggressive behavior with a median 5-year overall survival of 45% [5] and presence of lymph node metastases at first diagnosis in 70% of cases [36]. Additionally, the expression of a potential molecular target is of particular interest in metastatic lesions. Furthermore, five serial sections of the complete tumor (3 SaDu, 2 MuEp) were examined to evaluate the intratumoral staining heterogeneity. Eventually, two serial sections of non-cancerous salivary gland tissue were examined. A selection from the following tests to resolve unequivocal diagnoses in the cohort was used: immunohistochemical (IHC) staining for CK7, p63, NOR-1, SOX10, androgen receptor and HER2, FISH break-apart probes targeting MYB, MYBL1, PRKD1, PRKD2, PRKD3, EWSR1, MAML2, and ETV6 genes, as well as Sanger sequencing of PRKD1 hotspot mutations [37]. Tissue slides were stained with antibodies against CD138 (Cellmarque, clone: B-A38, host: Mouse, dilution: 1:100, pretreatment: EDTA). All IHC stainings were carried out with a Leica BOND-MAX stainer (Leica Biosystems, Wetzlar, Germany) in accordance with the manufacturer’s protocol. Counterstaining was done using haematoxylin and bluing reagent.

Two pathologists with special expertise in the field of SGC (CA, AQ) assessed the membranous CD138 expression for each tissue cylinder on the TMAs. The percentage of tumor cells with CD138 expression in relation to all tumor cells was calculated. A tumor cell was counted positive if ≥50% of the membrane showed CD138 expression. The final percentage of CD138-expressing tumor cells represents the mean value of the four cylinders per case.

### 4.3. MALDI-MS Imaging

For MALDI-MS imaging, analysis sections consecutive to immunohistochemistry-stained sections were used. Therefore, a representative sample of seven exemplary cases of primary SGC (two MuEp, two Acin, two ACC, and one SaDu) with immunohistochemical CD138 expression ranging between 9 and 48% of CD138 positive tumor cells and one CD138 negative secretory carcinoma (SeC) were analyzed to validate the immunohistochemical findings. Sample preparation, including deparaffinization, antigen retrieval, tryptic digestion, and matrix application, was performed according to a previously published study [38]. MALDI-MS imaging measurements were performed on an ultrafleXtreme mass spectrometer (Bruker Daltonik GmbH, Germany). Data acquisition was operated in reflective negative mode with 50 μm spatial resolution (medium laser spot size) and 200 laser shots. For data and imaging analysis, SCiLS Lab software Version 2021a Premium 3D (Bruker Daltonik GmbH, Germany) was used. In silico protein digestion was performed using the ProteinProspector MS-Digest tool (Uniprot ID: P18827). For the generation of the box plots, the mean intensity of all detected CD138 peptide masses per pixel were used. The final intensity represents the mean value for four cores per case. The expression of CD138 protein masses were evaluated with ROC analysis.

### 4.4. Statistical Analysis

Statistical analyses were performed using SPSS software version 28.0 (IBM, Armonk, New York, NY, USA). Distribution was tested using the Shapiro–Wilk test. The chi-squared test was used to evaluate if there was a statistically significant difference between the expected and the observed frequencies of two categorical variables. The Kruskal–Wallis test was used to determine if there was a statistically significant difference between more than two groups of an independent variable on a metric dependent variable. Spearman’s rank correlation coefficient was calculated for the correlation between non-normally distributed metric variables. The Kaplan–Meier method with 95% confidence intervals was used to test for progression-free survival (PFS) probability rates. In this context, the log-rank test was used for testing for statistical significance. PFS was defined as the time interval between the end of treatment and the date of progression of the specific disease or death. A *p*-value < 0.05 was considered statistically significant. R studio (version 2021.09.1) was used for visualization of box plots (ggplot2 package).

## Figures and Tables

**Figure 1 ijms-23-09037-f001:**
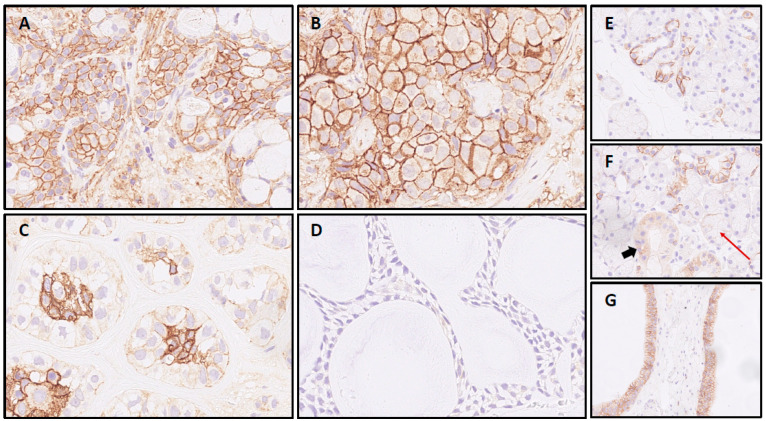
CD138 IHC in salivary gland carcinomas: Mucoepidermoid carcinoma (**A**) and salivary duct carcinoma sample (**B**), both with intense circumferential staining of >95% of cells. Epithelial-myoepithelial carcinoma (**C**) with signal restricted to luminal cells and adenoid cystic carcinoma case (**D**) with negativity for CD138; CD138 expression in normal submandibular gland tissue: (**E**) Most intercalated ducts with partial membranous staining in contrast to (**F**) striated ducts without membranous signal (cytoplasmatic positivity, bold arrow) and acini with negativity for CD138 (red arrow). (**G**) Two excretory ducts with basolateral SDC1 expression in >90% of duct epithelia.

**Figure 2 ijms-23-09037-f002:**
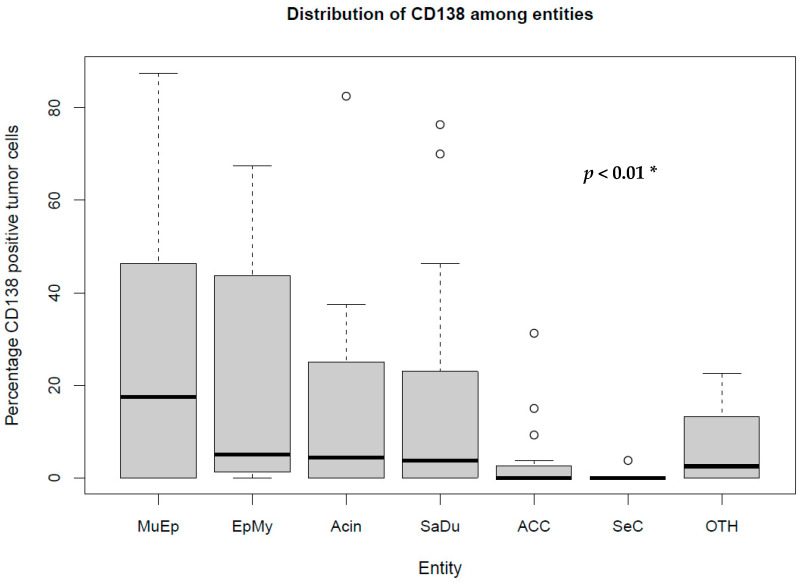
Box plot displaying the distribution of the percentages of CD138-expressing tumor cells among the most frequent entities. MuEp = Mucoepidermoid carcinoma, EpMy = Epithelial-myoepithelial carcinoma, Acin = Acinic cell carcinoma, SaDu = Salivary duct carcinoma, ACC = Adenoid cystic carcinoma, SeC = Secretory carcinoma, OTH = Others. * Kruskal–Wallis test. ◦ Outliers.

**Figure 3 ijms-23-09037-f003:**
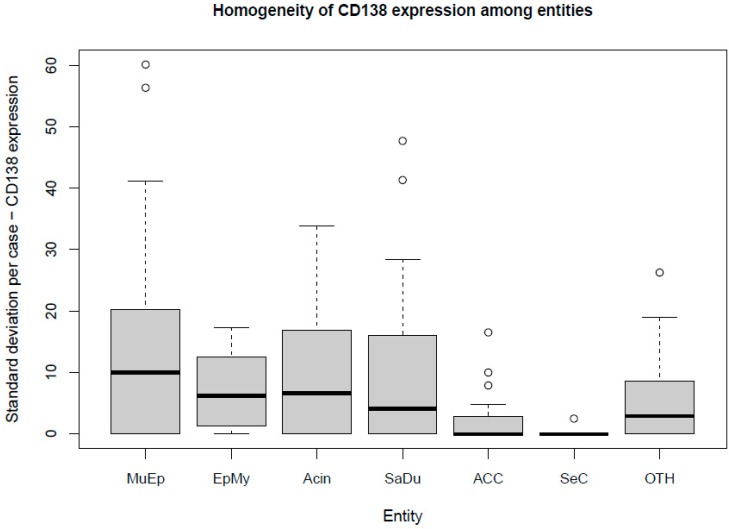
Box plot displaying the homogeneity of CD138 expression among the most frequent entities. MuEp = Mucoepidermoid carcinoma, EpMy = Epithelial-myoepithelial carcinoma, Acin = Acinic cell carcinoma, SaDu = Salivary duct carcinoma, ACC = Adenoid cystic carcinoma, SeC = Secretory carcinoma, OTH = Others. ◦ Outliers.

**Figure 4 ijms-23-09037-f004:**
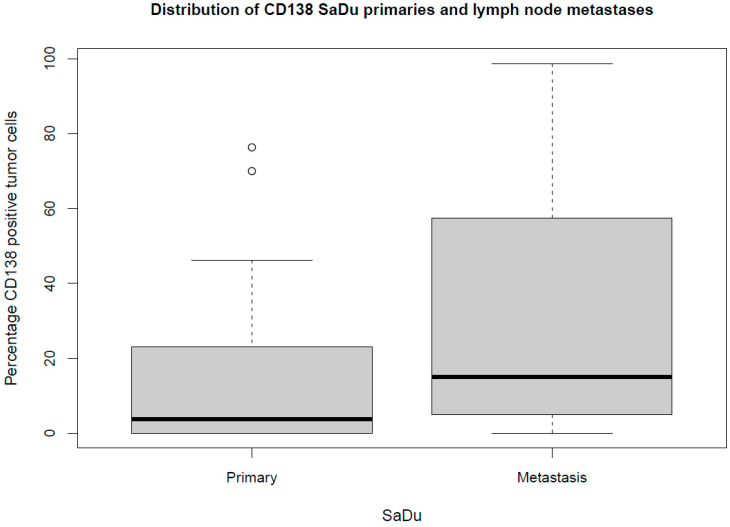
Box plot displaying the distribution of the percentages of CD138-expressing tumor cells for salivary duct carcinoma (SaDu) primaries and lymph node metastases. ◦ Outliers.

**Figure 5 ijms-23-09037-f005:**
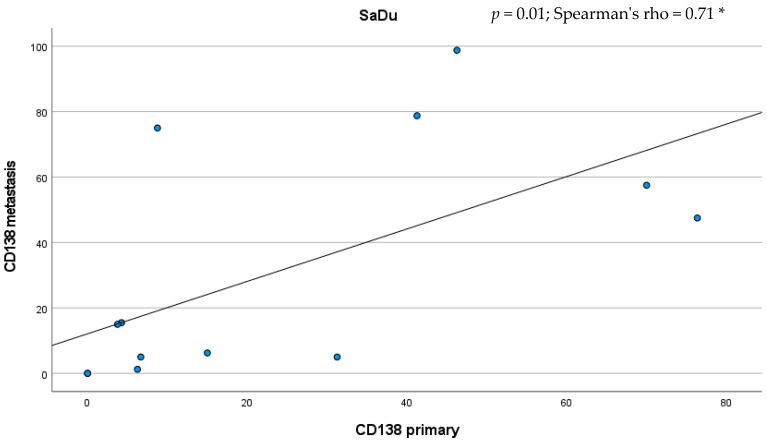
Scatter plot displaying the correlation between the percentage of CD138-expressing tumor cells in salivary duct carcinoma (SaDu) primaries and their lymph node metastases. * Spearman’s rank correlation coefficient.

**Figure 6 ijms-23-09037-f006:**
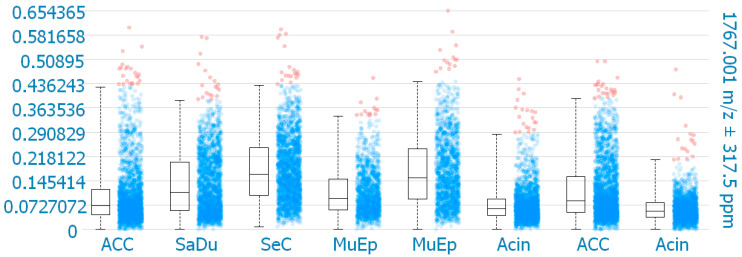
CD138 expression using MALDI-MS imaging. Box plots comparing the mean intensities of the *m*/*z* value of all detected CD138 peptide masses per pixel in the corresponding TMA cores. ACC = Adenoid cystic carcinoma, SaDu = Salivary duct carcinoma, SeC = Secretory carcinoma, MuEp = Mucoepidermoid carcinoma, Acin = Acinic cell carcinoma.

**Figure 7 ijms-23-09037-f007:**
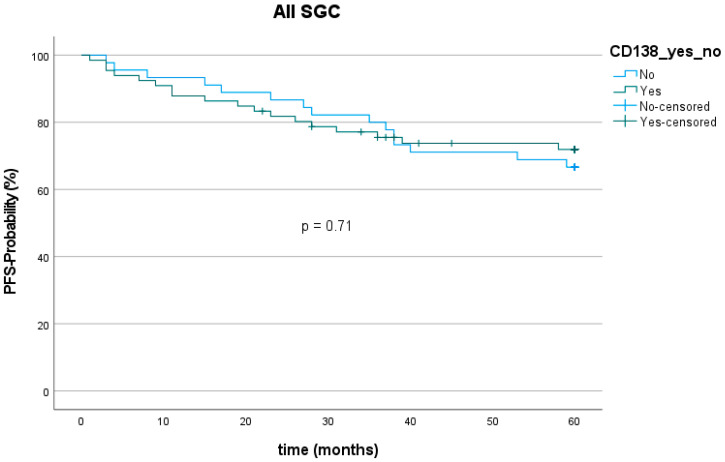
Kaplan–Meier curve and *p*-value of log-rank test for CD138 expression among all entities of salivary gland cancer (SGC). PFS = Progression-free survival.

**Table 1 ijms-23-09037-t001:** Localization of the primary tumor, demographic data, histopathological data, percentage of tumor cells with CD138 expression, and percentage of tumors with CD138 expression for the whole cohort and for the most frequent entities.

	All *n* = 111	SaDu *n* = 27	MuEp *n* = 25	ACC *n* = 23	Acin *n* = 12	EpMy *n* = 8	SecC *n* = 7	OTH *n* = 9
Localization								
Parotid gland	100 (90.1)	25 (92.6)	25 (100.0)	16 (69.6)	12 (100.0)	8 (100.0)	6 (85.7)	8 (88.9)
Submandibular gland	11 (9.9)	2 (7.4)	0 (0.0)	7 (30.4)	0 (0.0)	0 (0.0)	1(14.3)	1 (11.1)
Demographics								
Female	57 (51.4)	6 (22.2)	18 (72.0)	15 (65.2)	7 (58.3)	2 (25.0)	3 (42.99)	6 (66.7)
Male	54 (48.6)	21 (77.8)	7 (28.0)	8 (34.8)	5 (41.7)	6 (75.0)	4 (57.1)	3 (33.3)
Age	55.7 ± 17.6	67.2 ± 11.5	44.6 ± 18.3	51.7 ± 13.9	52.9 ± 18.6	62.6 ± 18.0	48.6 ± 20.0	66.1 ± 10.4
Histopathological parameters T-stage								
T1–2	55 (49.5)	10 (37.0)	17 (68.0)	10 (43.5)	6 (50.0)	5 (62.5)	5 (71.4)	2 (22.2)
T3–4	53 (47.7)	17 (63.0)	7 (28.0)	12 (52.2)	6 (50.0)	3 (37.5)	2 (28.6)	6 (66.7)
N/A	3 (2.7)	0 (0.0)	1 (4.0)	1 (4.3)	0 (0.0)	0 (0.0)	0 (0.0)	1 (11.1)
N-stage								
N0	73 (65.8)	4 (14.8)	21 (84.0)	16 (69.6)	10 (83.3)	8 (100.0)	6 (85.7)	8 (88.9)
N+	34 (30.6)	23 (85.2)	3 (12.0)	6 (26.1)	1 (8.3)	0 (0.0)	1 (14.3)	1 (11.1)
N/A	4 (3.6)	0 (0.0)	1 (4.0)	1 (4.3)	1 (8.3)	0 (0.0)	0 (0.0)	0 (0.0)
Vascular invasion								
V0	93 (83.8)	21 (77.8)	23 (92.0)	18 (78.3)	11 (91.7)	7 (87.5)	6 (85.7)	8 (88.9)
V1	10 (9.0)	6 (22.2)	2 (8.0)	0 (0.0)	0 (0.0)	1 (12.5)	1 (14.3)	0 (0.0)
N/A	8 (7.2)	0 (0.0)	0 (0.0)	5 (21.5)	1 (8.3)	0 (0.0)	0 (0.0)	1 (11.1)
Perineural invasion								
Pn0	61 (55.0)	5 (18.5)	22 (88.0)	8 (34.8)	9 (75.0)	7 (87.5)	6 (85.7)	4 (44.4)
Pn1	41 (36.9)	22 (81.5)	2 (8.0)	11 (47.8)	2 (16.7)	1 (12.5)	1 (14.3)	3 (33.3)
N/A	9 (8.1)	0 (0.0)	1 (4.0)	4 (17.4)	1 (8.3)	0 (0.0)	0 (0.0)	2 (22.2)
Lymphovascular invasion								
L0	89 (80.2)	15 (55.6)	24 (96.0)	18 (78.3)	10 (83.3)	7 (87.5)	7 (100.0)	8 (88.9)
L1	15 (13.5)	12 (44.4)	1 (4.0)	0 (0.0)	1 (8.3)	1 (12.5)	0 (0.0)	0 (0.0)
N/A	7 (6.3)	0 (0.0)	0 (0.0)	5 (21.5)	1 (8.3)	0 (0.0)	0 (0.0)	1 (11.1)
Extracapsular extension								
ECE−	85 (76.6)	13 (48.1)	24 (96.0)	15 (65.2)	10 (83.3)	8 (100.0)	7 (100.0)	8 (88.9)
ECE+	19 (17.1)	14 (51.9)	1 (4.0)	3 (13.0)	1 (8.3)	0 (0.0)	0 (0.0)	0 (0.0)
N/A	7 (6.3)	0 (0.0)	0 (0.0)	5 (21.7)	1 (8.3)	0 (0.0)	0 (0.0)	1 (11.1)
Grading								
Low	26 (23.4)	7 (25.9)	15 (60.0)	1 (4.3)	1 (8.3)	0 (0.0)	2 (28.6)	0 (0.0)
High/intermediate	51 (45.9)	17 (63.0)	10 (40.0)	16 (69.6)	2 (16.7)	1 (12.5)	1 (14.3)	4 (44.4)
N/A	34 (30.6)	3 (11.1)	0 (0.0)	6 (26.1)	9 (75.0)	7 (87.5.0)	4 (57.1)	5 (55.6)
CD138 (mean % of tumor cells, ±)	13.9 ± 21.8	15.2 ± 22.0	25.2 ± 26.7	3.0 ± 7.1	16.0 ± 24.9	20.9 ± 28.9	0.5 ± 1.4	8.1 ± 9.4
CD138 expression								
No	45 (40.5)	8 (29.6)	7 (28.0)	15 (65.2)	4 (33.3)	2 (25.0)	6 (85.7)	3 (33.3)
Yes	66 (59.5)	19 (70.4)	18 (72.0)	8 (34.8)	8 (66.7)	6 (75.0)	1 (14.3)	6 (66.7)
CD138 (mean % of tumor cells, ±, if CD138 positive)	23.4 ± 25.0	21.6 ± 23.5	35.0 ± 25.3	24.1 ± 27.4	16.0 ± 24.9	27.9 ± 30.6	3.8	12.1 ± 9.1

*n* = number of patients, ( ) percentages, ±standard deviation, MuEp = Mucoepidermoid carcinoma, EpMy = Epithelial-myoepithelial carcinoma, Acin = Acinic cell carcinoma, SaDu = Salivary duct carcinoma, ACC = Adenoid cystic carcinoma, SeC = Secretory carcinoma, OTH = Others.

**Table 2 ijms-23-09037-t002:** Statistical association between localization of the primary tumor, sex, histopathological data, and immunohistochemical CD138 expression (yes vs. no) for all entities and for the most frequent entities salivary duct carcinoma (SaDu), mucoepidermoid carcinoma (MuEp), and adenoid cystic carcinoma (ACC).

Variable	CD138 Expression (All Entities)		CD138 Expression (SaDu)		CD138 Expression (MuEp)		CD138 Expression (ACC)	
Parotid gland	60.0%	*p* = 0.73 #	68.0%	*p* = 0.34 #	72.0%	N/A	31.3%	*p* = 0.59 #
Submandibular gland	54.5%		100.0%		72.0%		42.9%	
Male	61.1%	*p* = 0.73 #	76.2%	*p* = 0.22 #	57.1%	*p* = 0.30 #	50.0%	*p* = 0.26 #
Female	57.9%		50.0%		77.8%		27.7%	
T1/2	67.3%	*p* = 0.13 #	80.0%	*p* = 0.40 #	76.5%	*p* = 0.80 #	50.0%	*p* = 0.23 #
T3/4	52.8%		64.7%		71.4%		25.0%	
N0	60.3%	*p* = 0.36 #	75.0%	*p* = 0.83 #	76.2%	*p* = 0.25 #	37.5%	*p* = 0.74 #
N+	61.8%		69.7%		66.7%		33.3%	
V0	59.1%	*p* = 0.58 #	71.4%	*p* = 0.82 #	73.9%	*p* = 0.47 #	27.8%	N/A
V1	70.0%		66.7%		50.0%		0.0%	
Pn0	60.7%	*p* = 0.70 #	60.0%	*p* = 0.57 #	72.7%	*p* = 0.50 #	37.5%	*p* = 0.64 #
Pn1	58.5%		72.7%		50.0%		27.3%	
L0	58.4%	*p* = 0.60 #	73.3%	*p* = 0.71 #	75.0%	*p* = 0.10 #	27.8%	N/A
L1	66.7%		66.7%		0.0%		0.0%	
ECE−	58.8%	*p* = 0.67 #	69.2%	*p* = 0.90 #	75.0%	*p* = 0.10 #	26.7%	*p* = 0.81 #
ECE+	63.2%		71.4%		0.0%		33.3%	
G1	69.2%	*p* = 0.13 #	58.8%	*p* = 0.20 #	70.0%	*p* = 0.86 #	0.0%	*p* = 0.57 #
G2/3	50.1%		85.7%		73.3%		25.0%	

Significance level *p* < 0.05, # chi-squared test.

## Data Availability

The data presented in this study are available on request from the corresponding author. The data are not publicly available due to privacy restrictions.

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
