# Peer review of "CD138 Is Expressed in Different Entities of Salivary Gland Cancer and Their Lymph Node Metastases and Therefore Represents a Potential Therapeutic Target"

_ijms, 2022, doi:10.3390/ijms23169037_

Round 1
Reviewer 1 Report
2.1 line 95-96. The authors state that the patients were phoned to follow up on current tumor status. Considering that cases back to 1990 were included, I wonder whether all subjects could be contacted. And when the authors were not able to contact a person, what did they do with their histological data? Were they excluded? Please clarify
2.2 line 120-121. How did the authors measure whether >50% of the membrane was positive? This please this information
Table 1, page 5: The stratification of MuEp into ECE-, ECE+ and N/A does not result in a total of N=25. Please verify the numbers and correct them
3.2 .line 177-178: Across all entities, a mean on only 13.9% of the tumor cells showed CD138 expression. This is a rather low number, and I think the authors should include this when the discuss the use of CD138 targeting drugs.
Figure 4, page 8: I think an X-Y scatter plot of the correlation between primary and metastates will be been more illustrative of the correlation
3.3 line 231-235. The authors present here results that deomonstrate that the CD138 expression observed by MALDI-MS does not correlate with the intensity of the membrane staining. I think this lack of correlation should be discussed in more detail, as it also relates to the usefulness of CD138 targeting drugs.
Page 9: Table 2 and the description of the results in the text above it are completely incomprehensible to me. What do these numbers refer to? To the MALDI data? But then I don't understand the stratification into subgroups? Or do they relate to the immunohistochemistry data? But in that case, why are the numbers different from Table 1. Please provide extensively more information on what this Table is based
Page 11, line 309-315 are just a mere repetition of the results, and can be removed from the Discussion
Reviewer 2 Report
Dear Authors, thank you very much for your paper. the study is important, well performed and well organized. Results are clearly described and analysed. Such studies are important as providing information for possible adjunctive/alternative therapies. I have only a suggestion, that is to widely discuss in the the introduction CD138 use for all salivary gland study you may found in literature, also in normal tissue, Sjogren Syndrome, etc etc, and not only as you performed that is a comparative for MM and other malignant neoplasm.
Thank you again for your paper
